# Predictive value of anthropometric measurements in survival and free walking ability of geriatric hip fractures after surgery

**Liqiang Wang, Zhibang Zhao, Wenliang Fan, Yuan Yao, Qingbo Chu** *

Emergency Trauma Center, Nanyang Second People's Hospital, Nanyang, Henan, China

* 13721809268@163.com

**Data Availability Statement:** The data used in this study were de-identified. The datasets used and/or analyzed during the current study are available in supplementary files.

## Abstract

### Background

We aimed to explore the predictive value of anthropometric measurements in survival and free walking ability of geriatric hip fractures after surgery.

### Methods

Eight common anthropometric measurements, including arm circumference (AC), waist circumference (WC), thigh circumference (TC), calf circumference (CC), biceps skinfold (BS), triceps skinfold (TS), suprailiac skinfold (SIS), and subscapular skinfold (SSS), were included to identify their predictive value in survival and free walking ability of geriatric hip fractures. The results of anthropometric measurements were compared between patients with different outcomes. Cox and logistics models were established to further identify the predictive value of anthropometric measurements.

### Results

Comparison among groups indicated that individuals with different outcomes may have significantly different anthropometric measurements. In the Cox analyses based on all individuals, all models proved that the patients with higher AC, as well as CC and BS, may have a lower risk of 1-year mortality. Similarly, in the logistics analysis, AC, CC, and BS were proven to have strong predictive ability for 6-month and 1-year mortality in females and overall individuals. However, the predictive value of the eight common anthropometric measurements in free walking ability is not significant.

### Conclusion

AC, CC, and BS may have strong predictive ability for 6-month and 1-year mortality in all individuals and females.

**Funding:** The author(s) received no specific funding for this work.

**Competing interests:** The authors have declared that no competing interests exist.

## Introduction

Hip fractures, occurring in the proximal femur, are commonly considered one of the most dangerous fractures in elder adults [1]. These fractures are primarily caused by low-energy trauma, often resulting from falls, while the root causes are age-related factors such as impaired balance, muscle weakness, and decreased bone mineral density [2]. Geriatric hip fractures are associated with high morbidity and mortality rates [3], primarily due to pre-existing comorbidities, surgical complications, and postoperative immobility [4]. Common complications include surgical site infections, urinary tract infections, venous thromboembolism, pressure ulcers, pneumonia, and delirium.

Nutrition status is a major risk factor for the outcomes of hip fractures and significantly impacts the rehabilitation of geriatric hip fractures [5]. Adequate nutrition is essential for bone regeneration, tissue repair, immune function, and wound healing, all of which may reduce the incidence of complications and improve the prognosis of geriatric hip fractures [6]. However, malnutrition as a common concern in older adults may contribute to poor healing, increased complications, prolonged recovery, low survival, and high disability following a hip fracture [7].

Assessing and addressing the nutritional needs of older adults with hip fractures is vital to improving their outcomes [5]. Anthropometric measurements, such as arm circumference, waist circumference, and others, can reflect an individual's nutritional status [8]. Poor anthropometric measurements are often associated with malnutrition, which can negatively impact the healing process and functional recovery in geriatric hip fracture patients. These anthropometric measurements provide valuable information about body composition and can assist healthcare professionals in assessing the nutritional needs of older adults with hip fractures. In this study, eight common anthropometric measurements, including arm circumference (AC), waist circumference (WC), thigh circumference (TC), calf circumference (CC), biceps skinfold (BS), triceps skinfold (TS), suprailiac skinfold (SIS), and subscapular skinfold (SSS), were included to identify their predictive value in survival and free walking ability of geriatric hip fractures.

## Methods

### Overall design

This research was a prospective observational study conducted at the Emergency Trauma Center of Nanyang Second People's Hospital. The study adhered to the principles outlined in the Declaration of Helsinki and received approval from the Ethics Committee of Nanyang Second People's Hospital (ID: 2013 Research Review No. 21). Patient privacy was rigorously protected, and written informed consent was obtained from all participants included in the study. The overall design of our study was summarized in Fig 1. All the patients with hip fractures in our department between January 2014 and January 2021 who met the following criteria were included in our study. The exclusion criteria in our study were: A. age younger than 50 years; B. fractures caused by high-energy injury; C. pathological fractures; D. no surgical procedures conducted; E. no available anthropometric measurements data; G: lost follow-up in the study procedures.

### Study data

Baseline characteristics were collected and summarized in this study, encompassing age, sex, body mass index (BMI), fracture type, fracture history, smoking history, alcoholism history, polytrauma, surgical procedures, anesthesia, and time from injury to surgery. Fracture types were categorized as either femoral neck fracture or intertrochanteric fracture, while surgical

**Fig 1. Overall design of our study.**

procedures were classified as internal fixation or arthroplasty. Furthermore, the first comprehensive hospital examination data after admission were collected, including electrocardiogram assessments, chest radiographs, red blood cell count (RBC), hemoglobin (Hb), blood glucose (GLU), and albumin (ALB) levels. The comorbidities statuses of patients were also recorded, and the Charlson comorbidity index (CCI) was used to evaluate the comorbidities of patients. CCI is a widely used tool in medical research and clinical practice to quantify the burden of comorbidities in patients. The index takes into account 19 different conditions, such as heart disease, diabetes, and cancer, and assigns a score ranging from 1 to 6 to each condition based on its severity and impact on mortality [9]. The items and their weights were summarized in S1 Table in S1 File.

## Anthropometric measurements

AC, WC, TC, and CC were measured by using a standard measuring tape, and the data of skinfold were obtained by using standard Lange skinfold calipers. To avoid the impact of fractures, the anthropometric measurements with two sides were measured on a side with no fractures. All measurements were conducted in triplicate by a single examiner, and the average value was used as the final result. The detailed measure procedures were: AC: measure the circumference at the midpoint of the upper arm; WC: measure the circumference at the at the midpoint between the lower rib margin and the iliac crest; TC: measure the circumference at the widest part of the thigh; CC: Wrap a measure the circumference at the widest part of the calf; BS: Pinch and measure the thickness of the skinfold on the front of the upper arm; TS: Pinch and measure the thickness of the skinfold on the back of the upper arm; SIS: Pinch and measure the thickness of the skinfold above the hip bone; SSS: Pinch and measure the thickness of the skinfold below the shoulder blade. Due to the hip fracture, all measurements were taken with the patient lying down.

## Follow-up and outcome

All enrolled participants in our study were monitored via telephone for a duration of 1 year. In our study, death was defined as any cause of mortality, while the ability to independently carry out daily activities without assistance was considered as free walking ability. The main objectives of the study were to assess mortality rates at 3 months, 6 months, and one-year post-

surgery, as well as evaluate the progress of free walking abilities at 3 months, 6 months, and one year.

## Statistical analyses

Continuous variables were presented as mean ± standard deviation and analyzed using Independent Student's T-tests for normally distributed data. For non-normally distributed data, Wilcoxon rank-sum tests were employed. Categorical variables were presented as count (percent) and evaluated using Chi-squared tests or Fisher's exact test. To meet the sample size of multivariable analyses, we calculated the minimum sample size by using (events per variable) EPV method [10]. Then the univariate Cox models were established and the significant variables were included in Cox models 1 and Cox models 2. Cox models 1 were adjusted for all types of anthropometric measurements, while Cox models 2 were adjusted for only one type of anthropometric measurements. To better identify the predictive value of anthropometric measurements for 3-month, 6-month, and 1-year mortality and walking ability, logistics analyses were also performed. Similar to Cox models, the logistics modes 1 and 2 were also established by adjusting for all types or single type of anthropometric measurements, respectively. ROC curves were established to evaluate the predictive values and to determine the potential cut-off values for each measurement. To avoid the bias caused by sex, the analyses were also performed in males and females separately. The statistical analyses were performed using R software version 4.2.2 (R Foundation for Statistical Computing, Vienna, Austria), and certain analyses were visualized using GraphPad Prism version 8.0.1 (GraphPad Software, San Diego, CA). A significance level of $P < 0.05$ was considered to reject the null hypothesis.

## Results

### Baseline features

Finally, a total of 544 individuals who underwent hip surgeries in our department between January 2017 and January 2021 were included in our study. In S2-S7 Tables in S1 File, the baseline features were summarized and compared between patients grouped by 3-month, 6-month, and 1-year mortality and walking ability. The mean age of the study population was 72.38 ± 10.47 years, and the mean BMI was 21.77 ± 4.27 kg/m$^2$. More female participants (374, 68.75%) were included in this study than males (170, 31.25%). Of all included patients, 95 of them died within one year after surgery. In comparing baseline characteristics, the patients with different outcomes have significantly different ages (S2-S7 Tables in S1 File).

### Anthropometric measurements in patients with different outcomes

The anthropometric measurements in patients grouped by 3-month, 6-month, and 1-year mortality and walking ability were summarized and compared. In all individuals (S8 Table in S1 File), compared with the patients who survived less than 3 months, the patients who survived more than 3 months have a significantly higher AC (27.55 vs. 24.97, p = 0.015), WC (86.79 vs. 84.48, p = 0.035), TC (42.50 vs. 39.83, p = 0.032), CC (30.76 vs. 24.99, p < 0.001), and SSS (13.01 vs 15.07, p < 0.001). Similarly, the patients who survived more than 6 months have a significantly higher AC (27.74 vs. 24.26, p < 0.001), TC (42.67 vs. 39.48, p < 0.001), CC (31.08 vs. 24.75, p < 0.001), and BS (12.99 vs 11.88, p = 0.001) than those who have a survival less than 6 months. Moreover, the patients who survived more than 1 year have a significantly higher AC (27.72 vs. 26.33, p = 0.002), CC (30.93 vs. 29.05, p = 0.001), and BS (13.03 vs 12.30, p = 0.002). In the analyses for patients with different free walking abilities, the patients with free walking ability at 1 year have a significantly higher AC (27.74 vs. 26.71, p = 0.010), and CC

**Table 1. Comparison of anthropometric measurements in patients with different outcomes (males).**

|  | Survival > 3 months N = 165 | Survival ≤ 3 months N = 5 | p | Free walking > 3 months N = 128 | Free walking ≤ 3 months N = 42 | p |
|---|---|---|---|---|---|---|
| AC | 27.61 (3.95) | 25.15 (4.69) | 0.174 | 27.53 (3.96) | 27.54 (4.08) | 0.986 |
| WC | 87.27 (4.31) | 83.62 (3.84) | 0.063 | 86.89 (4.10) | 88.02 (4.92) | 0.14 |
| TC | 42.51 (4.95) | 42.47 (4.57) | 0.872 | 42.16 (5.01) | 43.56 (4.57) | 0.076 |
| CC | 31.69 (5.19) | 24.38 (7.28) | 0.003 | 31.34 (5.62) | 31.91 (4.61) | 0.553 |
| BS | 13.04 (2.10) | 12.22 (2.29) | 0.392 | 13.00 (2.07) | 13.06 (2.23) | 0.881 |
| TS | 15.19 (2.48) | 15.72 (1.98) | 0.635 | 15.29 (2.46) | 14.96 (2.49) | 0.454 |
| SIS | 16.30 (2.54) | 16.50 (2.64) | 0.867 | 16.18 (2.43) | 16.70 (2.82) | 0.254 |
| SSS | 13.35 (2.18) | 14.69 (0.96) | 0.171 | 13.26 (2.18) | 13.77 (2.10) | 0.187 |
|  | Survival > 6 months N = 158 | Survival ≤ 6 months N = 12 | p | Free walking > 6 months N = 80 | Free walking ≤ 6 months N = 90 | p |
| AC | 27.87 (3.77) | 23.17 (4.35) | <0.001 | 27.06 (4.07) | 27.96 (3.88) | 0.144 |
| WC | 87.32 (4.37) | 85.09 (3.27) | 0.085 | 86.71 (4.10) | 87.58 (4.51) | 0.192 |
| TC | 42.75 (4.89) | 39.37 (4.46) | 0.015 | 42.32 (4.96) | 42.67 (4.92) | 0.626 |
| CC | 31.93 (5.12) | 25.54 (5.41) | <0.001 | 30.98 (6.03) | 31.92 (4.72) | 0.256 |
| BS | 13.13 (2.01) | 11.56 (2.77) | 0.013 | 12.94 (2.17) | 13.09 (2.05) | 0.646 |
| TS | 15.18 (2.47) | 15.53 (2.44) | 0.642 | 15.23 (2.24) | 15.19 (2.67) | 0.913 |
| SIS | 16.29 (2.55) | 16.56 (2.48) | 0.72 | 16.13 (2.28) | 16.46 (2.74) | 0.403 |
| SSS | 13.42 (2.13) | 12.98 (2.61) | 0.504 | 13.28 (2.15) | 13.48 (2.18) | 0.536 |
|  | Survival > 1 year N = 136 | Survival ≤ 1 year N = 34 | p | Free walking > 1 year N = 46 | Free walking ≤ 1 year N = 124 | p |
| AC | 27.70 (3.49) | 26.86 (5.54) | 0.27 | 26.88 (4.56) | 27.78 (3.73) | 0.196 |
| WC | 87.28 (4.31) | 86.73 (4.45) | 0.511 | 86.21 (3.94) | 87.52 (4.43) | 0.078 |
| TC | 42.58 (4.86) | 42.22 (5.24) | 0.708 | 42.49 (4.97) | 42.52 (4.93) | 0.936 |
| CC | 31.76 (4.82) | 30.35 (7.17) | 0.173 | 29.46 (6.27) | 32.23 (4.82) | 0.003 |
| BS | 13.16 (1.95) | 12.44 (2.58) | 0.076 | 12.87 (2.43) | 13.07 (1.98) | 0.593 |
| TS | 15.28 (2.50) | 14.94 (2.33) | 0.476 | 15.16 (2.24) | 15.23 (2.55) | 0.868 |
| SIS | 16.40 (2.54) | 15.95 (2.54) | 0.357 | 16.01 (2.51) | 16.42 (2.55) | 0.346 |
| SSS | 13.45 (2.09) | 13.13 (2.45) | 0.439 | 13.02 (2.31) | 13.52 (2.10) | 0.183 |

Note: AC: arm circumference, WC: waist circumference, TC: thigh circumference, CC: calf circumference, BS: biceps skinfold, TS: triceps skinfold, SIS: suprailiac skinfold, SSS: subscapular skinfold.

(30.92 vs. 29.67, p = 0.015). However, there were no significant differences in other anthropometric measurements.

In the males (Table 1), the AC was significantly different in patients grouped by 6-month survival (27.87 vs. 23.17, p < 0.001), as well as TC (42.75 vs. 39.37, p = 0.015), CC (31.93 vs. 25.54, p < 0.001), and BS (13.13 vs 11.56, p = 0.013). Moreover, compared with the patients who survived less than 3 months, the patients who survived more than 3 months have a significantly higher CC (31.69 vs. 24.38, p = 0.003).

In the females (Table 2), compared with the patients who survived less than 3 months, the patients who survived more than 3 months have a significantly higher AC (27.53 vs. 24.88, p = 0.045), TC (42.50 vs. 38.50, p = 0.012), CC (30.34 vs. 25.30, p = 0.002), SIS (15.96 vs 17.61, p = 0.036), and SSS (12.86 vs 15.27, p < 0.001). Similarly, the patients who survived more than 6 months have a significantly higher AC (27.69 vs. 24.71, p < 0.001), TC (42.63 vs. 39.52, p = 0.003), CC (30.69 vs. 24.43, p < 0.001), BS (12.92 vs 12.01, p = 0.025), and TS (15.06 vs. 13.85, p = 0.020) than those who have a survival less than 6 months. Moreover, the patients who survived more than 1 year have a significantly higher AC (27.73 vs. 26.04, p = 0.003), CC

**Table 2. Comparison of anthropometric measurements in patients with different outcomes (females).**

|  | Survival > 3 months N = 364 | Survival ≤ 3 months N = 10 | p | Free walking > 3 months N = 288 | Free walking ≤ 3 months N = 86 | p |
|---|---|---|---|---|---|---|
| AC | 27.53 (4.13) | 24.88 (3.16) | 0.045 | 27.44 (4.08) | 27.52 (4.31) | 0.863 |
| WC | 86.56 (4.11) | 84.91 (3.88) | 0.209 | 86.64 (4.00) | 86.11 (4.46) | 0.293 |
| TC | 42.50 (5.52) | 38.50 (4.58) | 0.012 | 42.39 (5.82) | 42.38 (4.44) | 0.803 |
| CC | 30.34 (5.10) | 25.30 (5.53) | 0.002 | 30.05 (5.20) | 30.72 (5.06) | 0.296 |
| BS | 12.82 (2.10) | 13.82 (2.14) | 0.14 | 12.83 (2.04) | 12.93 (2.31) | 0.692 |
| TS | 14.99 (2.66) | 14.05 (4.07) | 0.277 | 14.92 (2.72) | 15.13 (2.67) | 0.518 |
| SIS | 15.96 (2.45) | 17.61 (2.15) | 0.036 | 15.98 (2.43) | 16.11 (2.55) | 0.665 |
| SSS | 12.86 (2.05) | 15.27 (1.75) | <0.001 | 12.99 (2.11) | 12.71 (1.98) | 0.272 |
|  | Survival > 6 months N = 345 | Survival ≤ 6 months N = 29 | p | Free walking > 6 months N = 167 | Free walking ≤ 6 months N = 207 | p |
| AC | 27.69 (4.07) | 24.71 (3.89) | <0.001 | 27.33 (3.90) | 27.56 (4.31) | 0.593 |
| WC | 86.59 (4.04) | 85.69 (4.82) | 0.255 | 86.80 (4.25) | 86.30 (3.99) | 0.245 |
| TC | 42.63 (5.47) | 39.52 (5.44) | 0.003 | 42.40 (5.03) | 42.39 (5.91) | 0.658 |
| CC | 30.69 (4.80) | 24.43 (5.91) | <0.001 | 30.05 (5.43) | 30.33 (4.95) | 0.596 |
| BS | 12.92 (2.03) | 12.01 (2.77) | 0.025 | 12.82 (2.22) | 12.88 (2.02) | 0.799 |
| TS | 15.06 (2.65) | 13.85 (3.12) | 0.02 | 15.08 (2.84) | 14.88 (2.59) | 0.463 |
| SIS | 16.01 (2.39) | 15.95 (3.14) | 0.902 | 15.94 (2.49) | 16.06 (2.43) | 0.633 |
| SSS | 12.87 (2.00) | 13.55 (2.82) | 0.092 | 12.92 (2.27) | 12.93 (1.92) | 0.981 |
|  | Survival > 1 year N = 313 | Survival ≤ 1 year N = 61 | p | Free walking > 1 year N = 92 | Free walking ≤ 1 year N = 282 | p |
| AC | 27.73 (4.11) | 26.04 (3.97) | 0.003 | 26.62 (3.88) | 27.73 (4.18) | 0.025 |
| WC | 86.65 (4.07) | 85.88 (4.30) | 0.182 | 86.73 (4.21) | 86.45 (4.08) | 0.57 |
| TC | 42.47 (5.53) | 41.97 (5.57) | 0.53 | 42.67 (5.79) | 42.30 (5.45) | 0.742 |
| CC | 30.57 (4.81) | 28.33 (6.41) | 0.002 | 29.77 (5.92) | 30.35 (4.90) | 0.351 |
| BS | 12.97 (2.06) | 12.22 (2.24) | 0.01 | 12.73 (2.40) | 12.89 (2.00) | 0.536 |
| TS | 15.07 (2.67) | 14.42 (2.80) | 0.086 | 14.62 (3.02) | 15.08 (2.59) | 0.154 |
| SIS | 16.08 (2.36) | 15.65 (2.87) | 0.207 | 15.79 (2.72) | 16.08 (2.36) | 0.318 |
| SSS | 12.86 (2.01) | 13.29 (2.36) | 0.136 | 13.01 (2.57) | 12.90 (1.89) | 0.646 |

Note: AC: arm circumference, WC: waist circumference, TC: thigh circumference, CC: calf circumference, BS: biceps skinfold, TS: triceps skinfold, SIS: suprailiac skinfold, SSS: subscapular skinfold.

(30.57 vs. 28.33, p = 0.002), and BS (12.97 vs 12.22, p = 0.010). In the analyses for patients with different free walking abilities, the patients with free walking ability at 1 year have a significantly higher AC (27.73 vs. 26.51, p = 0.025). There were no significant differences in other anthropometric measurements.

## Results of multivariable analyses

Cox models were established to identify the risk roles of anthropometric measurements in 1-year mortality (Fig 2 and S9 Table in S1 File). Age, sex (in models for all individuals only), BMI, and fracture type were included in the Cox models 1 and 2 as co-variables. In the Cox analyses based on all individuals, all models proved that the patients with higher AC, as well as CC and BS, may have a lower risk of 1-year mortality. However, in the analyses based on males, only Model 1 and Model 2 indicated that individuals with higher CC and BS may face better 1-year survival. For females, the results were consistent with the models based on all individuals.

**Fig 2. Cox analyses for 1-year mortality for males and females.** Models 1 were adjusted for all types of anthropometric measurements; Models 2 were adjusted for only one type of anthropometric measurements. AC: arm circumference, WC: waist circumference, TC: thigh circumference, CC: calf circumference, BS: biceps skinfold, TS: triceps skinfold, SIS: suprailiac skinfold, SSS: subscapular skinfold.

The results of logistics analyses for 3-month, 6-month, and 1-year mortality and walking ability were summarized in Fig 3 and S10-S12 Tables in S1 File. Age, sex (in models for all individuals only), BMI, and fracture type were included in the multivariable logistics models 1 and 2 as co-variables. In the logistics analyses for 1-year mortality based on all individuals and females, the patients with higher AC, CC, and BS may have lower 1-year mortality, while the predictive value of AC, CC, and BS in males was not significant. For 1-year free walking ability, higher AC was proved to be a protective factor in the models based on all individuals, and lower CC might be a risk factor in the models based on males (Fig 3). Similarly, in the models

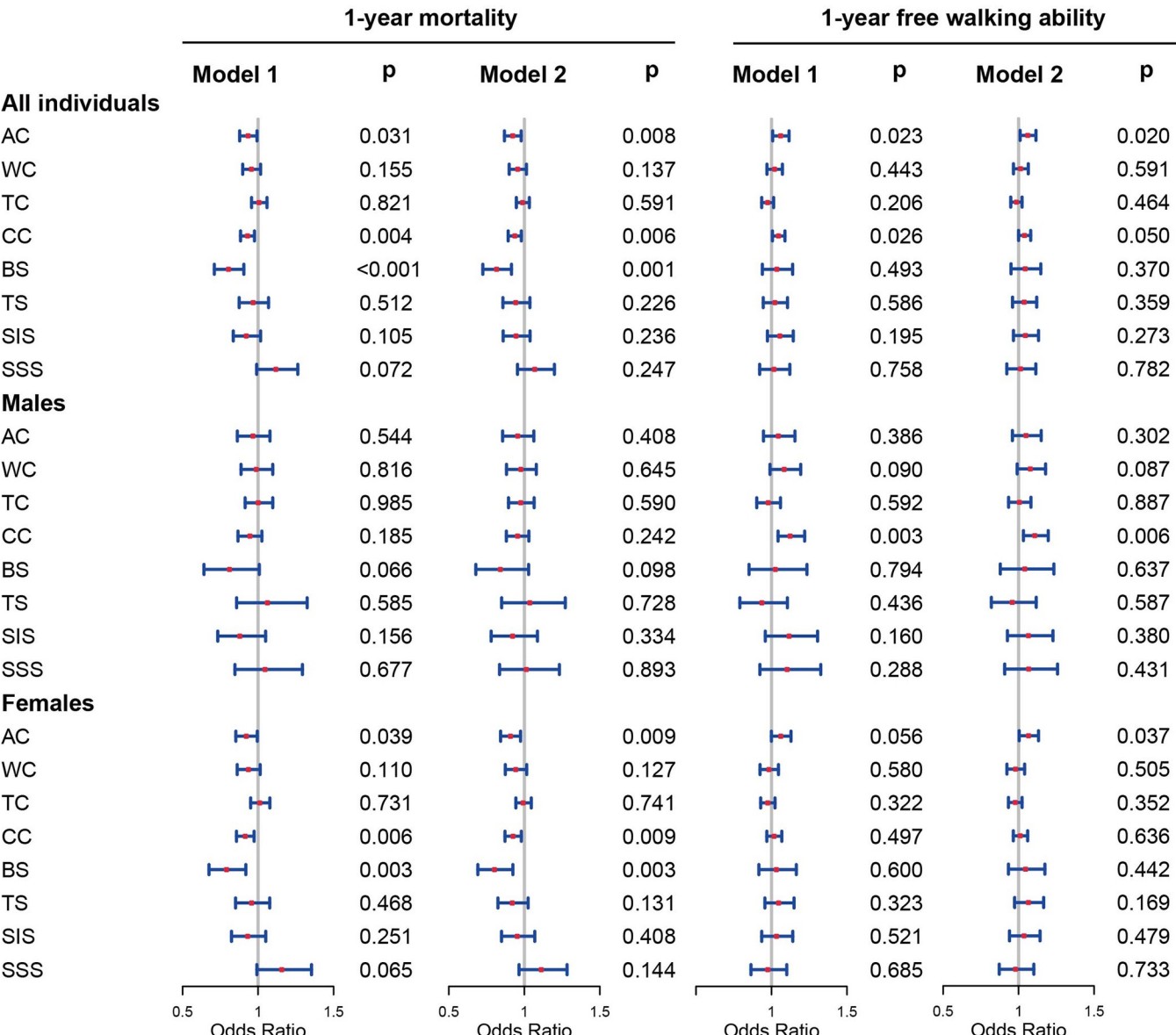

**Fig 3. Logistics analyses for 1-year mortality for males and females.** Models 1 were adjusted for all types of anthropometric measurements; Models 2 were adjusted for only one type of anthropometric measurements. AC: arm circumference, WC: waist circumference, TC: thigh circumference, CC: calf circumference, BS: biceps skinfold, TS: triceps skinfold, SIS: suprailiac skinfold, SSS: subscapular skinfold.

based on males, only a few anthropometric measurements were proven to be predictive factors in 3-month and 6-month mortality and walking ability. Moreover, for models for 6-month mortality, higher AC, CC, and BS were also proved to be protective factors in both all individuals and females (S10 and 12 Tables in S1 File).

## Predictive values of anthropometric measurements

ROC curves were established to determine the predictive values and the potential cutoff values of anthropometric measurements for mortality and free walking ability. The multivariable analysis indicated the predictive ability of AC, CC, and BS, so we summarized their detailed

**Table 3. Results of ROC analysis for AC, CC, and BS.**

| | Males | | | | | |
|---|---|---|---|---|---|---|
| | 1-year mortality | | | 1-year free walking ability | | |
| | AUC(95% CI) | Cutoff values | P | AUC(95% CI) | Cutoff values | p |
| AC | 0.556 (0.426, 0.687) | 23.25 | 0.156 | 0.556 (0.451, 0.661) | 26.295 | 0.87 |
| CC | 0.581 (0.459, 0.702) | 29.59 | 0.074 | 0.664 (0.564, 0.763) | 29.65 | 0.001 |
| BS | 0.600 (0.488, 0.712) | 13.01 | 0.964 | 0.539 (0.440, 0.639) | 13.165 | 0.786 |
| | Females | | | | | |
| | 1-year mortality | | | 1-year free walking ability | | |
| | AUC(95% CI) | Cutoff values | p | AUC(95% CI) | Cutoff values | p |
| AC | 0.617 (0.542, 0.691) | 28.245 | 0.002 | 0.575 (0.510, 0.640) | 28.31 | 0.015 |
| CC | 0.608 (0.523, 0.694) | 29.395 | 0.004 | 0.524 (0.455, 0.594) | 23.19 | 0.76 |
| BS | 0.589 (0.508, 0.670) | 12.98 | 0.014 | 0.517 (0.445, 0.589) | 12.98 | 0.686 |

Note: AC: arm circumference, CC: calf circumference, BS: biceps skinfold.

results of ROC curves in Table 3. The detailed results of ROC analyses for all anthropometric measurements, including AUC, 95%CI, potential cutoff values, and p values, were summarized in S13 Table in S1 File. Similarly, the AC, CC, and BS showed significant predictive values for 1-year mortality in all individuals and females, while no significant results in males. For 1-year free walking ability, AC was proven to have significant predictive ability in all individuals and females, and CC was only proven in all individuals, while no significant results of BS were proven. The potential cutoff values of each anthropometric measurement have been summarized in Table 3 and S13 Table in S1 File.

## Discussion

In this study, 8 common anthropometric measurements were included to identify their predictive value in the survival and free-walking ability of patients with hip fractures after surgery. Based on the comprehensive analysis of both univariate and multivariate analyses, we conclude that the predictive value of the eight common anthropometric measurements in elderly male patients after hip fracture surgery may not be significant in terms of mortality and rate of free walking. Additionally, their predictive role in free walking ability is also not significant in the both overall population and females. However, in all individuals and females, AC, CC, and BS have strong predictive ability for 6-month and 1-year mortality.

The relationship between AC and nutritional status in elderly individuals has been widely studied [11]. Low AC has been associated with malnutrition, muscle wasting, and frailty in older adults [12]. Individuals with lower arm circumference are more likely to experience poor nutritional status and have an increased risk of complications and mortality following hip fractures. In our study, the data also proved that patients with lower AC may have a higher risk of mortality after hip fracture. Therefore, arm circumference can serve as a valuable tool in assessing nutritional status and predicting prognosis in older adults with hip fractures. Low AC may also relate to sarcopenia and lower bone mass, which may also contribute to the poor outcomes of hip fractures [12, 13]. In a study focusing on community-dwelling older adults at risk of malnutrition, there is a significant and prevalent association between sarcopenia and lower calf circumference [14].

Calf circumference is an anthropometric measurement that has been utilized to assess the nutritional status of older adults and potentially predict the prognosis of hip fractures in this population [15]. Lower CC has been associated with poor nutritional status and in this study,

has been identified as a risk factor for adverse outcomes following hip fractures, including increased mortality rates [16]. Furthermore, CC may also serve as an indicator of sarcopenia, a condition characterized by muscle loss, which can further impact the prognosis of hip fractures in older adults [17]. Therefore, evaluating CC can provide valuable insights into both nutritional status and sarcopenia, enabling a comprehensive assessment of the prognosis of hip fractures in the elderly. In the previous study, the CC has been recommended to be used in multiple nutrition evaluation tools, including Subjective Global Assessment (SGA), the Mini Nutritional Assessment Long Form (MNA-LF), and the Global Leadership Initiative on Malnutrition (GLIM), the tools integrated with CC may have much more evaluation values [18]. Moreover, the predictive value of CC in elder adults for other diseases and healthy conditions was also proven, including cancer [19], cardiovascular diseases [20], and so on.

BS measurement, as an anthropometric parameter, holds promise in assessing the nutritional status of older adults and potentially predicting the prognosis of hip fractures in this population [21]. In our study, lower BS thickness has been identified as a significant risk factor for low survival rates following hip fractures. This association suggests that BS measurement can effectively reflect the nutritional status of older adults, which in turn impacts the recovery and prognosis of hip fractures [22]. Furthermore, considering the relationship between biceps skinfold thickness and sarcopenia, biceps skinfold measurement may also serve as a valuable predictor of hip fracture prognosis by indirectly reflecting the presence and severity of sarcopenia and muscle mass [23]. Therefore, incorporating biceps skinfold measurement into the comprehensive assessment of nutritional status and sarcopenia can provide valuable insights into the prognosis of hip fractures in the elderly population.

Our study is subject to several limitations that should be acknowledged. Firstly, as an observational study, this study had data loss and loss to follow-up, which may introduce bias into our findings. However, we have tried our best to control the sample size in this study. Assuming that loss to follow-up is a random event, if the sample size is sufficient, the patients who are not lost to follow-up can adequately represent all patients. We believe that the bias introduced by loss to follow-up should also be controllable. Secondly, the anthropometric measurement data were obtained when the patients were admitted to our department, and due to the limitation of hip fracture, all the anthropometric measurements were performed on the patients with forced postures. Therefore, our data may not be consistent with standard tests. However, all the patients were subjected to a standard measurement procedure, so that the bias caused by posture might be ignored. Thirdly, being an observational study, we were unable to establish a causal relationship between anthropometric measurements and fracture prognosis. Lastly, while we included the most relevant confounding factors in our multivariable models, certain variables such as the use of lipid-lowering drugs were not available, which may introduce bias into our results.

## Conclusion

The predictive value of the eight common anthropometric measurements in elderly male patients after hip fracture surgery may not be significant in terms of mortality and rate of free walking. Additionally, their predictive role in free walking ability is also not significant in the both overall population and females. However, in all individuals and females, AC, CC, and BS have strong predictive ability for 6-month and 1-year mortality.

## Supporting information

**S1 File. S1 Table: Detailed items and scores in CCI; S2 Table: Baseline chart of patients grouped by 3-month survival; S3 Table: Baseline chart of patients grouped by 3-month walking ability; S4 Table: Baseline chart of patients grouped by 6-month survival; S5**

Table: Baseline chart of patients grouped by 6-month walking ability; S6 Table: Baseline chart of patients grouped by 1-year survival; S7 Table: Baseline chart of patients grouped by 1-year walking ability; S8 Table: Comparison of anthropometric measurements in patients with different outcomes in all individuals; S9 Table: Detailed information of Cox analyses; S10 Table: Detailed information of logistics analyses based on all individuals; S11 Table: Detailed information of logistics analyses based on males; S12 Table: Detailed information of logistics analyses based on females; S13 Table: Detailed information of ROC curves; S14 Table: Raw data of our study.

(XLSX)

## Author Contributions

**Conceptualization:** Liqiang Wang.

**Data curation:** Liqiang Wang, Wenliang Fan.

**Investigation:** Zhibang Zhao.

**Methodology:** Zhibang Zhao, Wenliang Fan.

**Project administration:** Zhibang Zhao.

**Resources:** Zhibang Zhao, Yuan Yao.

**Software:** Yuan Yao.

**Supervision:** Yuan Yao.

**Writing – original draft:** Qingbo Chu.

**Writing – review & editing:** Qingbo Chu.

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
