## [Decision Letter · Decision Letter 0]

7 Nov 2023

PONE-D-23-24968Predictive value of anthropometric measurements in survival and free walking ability of geriatric hip fractures after surgery.PLOS ONE

Dear Dr. Chu,

Thank you for submitting your manuscript to PLOS ONE. After careful consideration, we feel that it has merit but does not fully meet PLOS ONE’s publication criteria as it currently stands. Therefore, we invite you to submit a revised version of the manuscript that addresses the points raised during the review process.

Please provide feedback on how your calculated your sample size and include location of your study.

Please submit your revised manuscript by Dec 22 2023 11:59PM.  If you will need more time than this to complete your revisions, please reply to this message or contact the journal office at plosone@plos.org. Please include the following items when submitting your revised manuscript:

We look forward to receiving your revised manuscript.

Kind regards,

Victor Afamefuna Egwuonwu, PhD

Academic Editor

PLOS ONE

Journal Requirements:

4. Please ensure that you refer to Figure 1 in your text as, if accepted, production will need this reference to link the reader to the figure.

**Additional Editor Comments:**

Dear Authors,

Please attend to reviewer one concern as quickly as you can and send modification for final consideration.

Thank you and best wishes

Reviewers' comments:

Reviewer's Responses to Questions

**Comments to the Author**

1. Is the manuscript technically sound, and do the data support the conclusions?

Reviewer #1: Partly

Reviewer #2: Yes

2. Has the statistical analysis been performed appropriately and rigorously? 

Reviewer #1: Yes

Reviewer #2: Yes

3. Have the authors made all data underlying the findings in their manuscript fully available?

Reviewer #1: No

Reviewer #2: Yes

4. Is the manuscript presented in an intelligible fashion and written in standard English?

Reviewer #1: Yes

Reviewer #2: Yes

5. Review Comments to the Author

Reviewer #1: The article expressed original thoughts with most of the various subsections involved in scientific work given. However, the sample size used for the research was missing including the location of the study otherwise a good article.

Reviewer #2: The manuscript was well-written however, I have one question regarding anthropometric measurements choices utilised in your research they did not include central obesity measure will this not have affected your outcome and does it mean that all your subjects have no obese individual from the start to finish the observational study.

6. PLOS authors have the option to publish the peer review history of their article (what does this mean?). If published, this will include your full peer review and any attached files.

Reviewer #1: **Yes: **Dr Ulu O. Ulu

Reviewer #2: No

---

## [Author Response · Author response to Decision Letter 0]

16 Nov 2023

Responses to editors

Thank you for your comments and tips concerning our manuscript. We would respond to reviewers and submit the revised manuscripts according to your prompts. Thank you. 

According to the journal requirements, we have edited our manuscript to meet PLOS ONE’s style requirements, and the ORCID would be updated in EM system. Moreover, the duplicated ethics statement has been deleted in our revised manuscript, and the reference to Figure 1 was also added. Lastly, the reference list has also been checked. 

Responses to reviewer 1

The article expressed original thoughts with most of the various subsections involved in scientific work given. However, the sample size used for the research was missing including the location of the study otherwise a good article.

Response:

Thank you for your kind comments. Our study included 544 individuals, and this study was conducted at the Emergency Trauma Center of Nanyang Second People’s Hospital. Due to the nature of observational study, we may not control the sample size accurately after following up. Therefore, we set a minimum sample size by calculating the minimum sample size required for multivariable regression. This study included 8 anthropometric measurements variables, and we calculated the sample size by (events per variable) EPV method. 

EPV = number of events / number of variables

When the EPV > 10, the multivariable models could be considered being stable. Therefore, the sample size was least 80 for events group [1]. 

Moreover, according to the similar studies in hip fractures, the sample size of our study (544 individuals) was also satisfactory [2]. 

More discussion of sample size and location of this study have added to our study (Method and Result part).

[1] Vittinghoff E, McCulloch CE. Relaxing the rule of ten events per variable in logistic and Cox regression. Am J Epidemiol. 2007 Mar 15;165(6):710-8. doi: 10.1093/aje/kwk052. Epub 2006 Dec 20. PMID: 17182981.

[2] Liu M, Chu Q, Yang C, Wang J, Fu M, Zhang Z, Sun G. The paradoxical relation between serum uric acid and outcomes of hip fracture in older patients after surgery: A 1-year follow-up study. Surgery. 2022 Nov;172(5):1576-1583. doi: 10.1016/j.surg.2022.07.008. Epub 2022 Aug 26. PMID: 36031447.

Responses to reviewer 2

The manuscript was well-written however, I have one question regarding anthropometric measurements choices utilised in your research they did not include central obesity measure will this not have affected your outcome and does it mean that all your subjects have no obese individual from the start to finish the observational study.

Response:

Thank you for your valuable advice. We have to admit that we cannot included all central obesity measures in our study. However, we believe some of measurements may be used as central obesity measures, such as waist circumference, thigh circumference, and suprailiac skinfold. Moreover, many studies used waist circumference as an index of central fatness [1]. Therefore, we believe the waist circumference in our study may be considered as a central fatness measure. 

[1] Jayedi A, Soltani S, Zargar MS, Khan TA, Shab-Bidar S. Central fatness and risk of all cause mortality: systematic review and dose-response meta-analysis of 72 prospective cohort studies. BMJ. 2020 Sep 23;370:m3324. doi: 10.1136/bmj.m3324. PMID: 32967840; PMCID: PMC7509947.

---

## [Decision Letter · Decision Letter 1]

14 Dec 2023

PONE-D-23-24968R1Predictive value of anthropometric measurements in survival and free walking ability of geriatric hip fractures after surgery.PLOS ONE

Dear Dr. Chu,

Thank you for submitting your manuscript to PLOS ONE. After careful consideration, we feel that it has merit but does not fully meet PLOS ONE’s publication criteria as it currently stands. Therefore, we invite you to submit a revised version of the manuscript that addresses the points raised during the review process.

We look forward to receiving your revised manuscript.

Kind regards,

Amir Human Hoveidaei

Academic Editor

PLOS ONE

Additional Editor Comments (if provided):

Dear authors,

Please follow the reviewer comments and revise the paper for a re-evaluation.

Reviewers' comments:

Reviewer's Responses to Questions

**Comments to the Author**

1. If the authors have adequately addressed your comments raised in a previous round of review and you feel that this manuscript is now acceptable for publication, you may indicate that here to bypass the “Comments to the Author” section, enter your conflict of interest statement in the “Confidential to Editor” section, and submit your "Accept" recommendation.

Reviewer #1: All comments have been addressed

Reviewer #2: All comments have been addressed

Reviewer #3: (No Response)

2. Is the manuscript technically sound, and do the data support the conclusions?

Reviewer #1: Yes

Reviewer #2: Yes

Reviewer #3: No

3. Has the statistical analysis been performed appropriately and rigorously? 

Reviewer #1: Yes

Reviewer #2: Yes

Reviewer #3: No

4. Have the authors made all data underlying the findings in their manuscript fully available?

Reviewer #1: Yes

Reviewer #2: Yes

Reviewer #3: Yes

5. Is the manuscript presented in an intelligible fashion and written in standard English?

Reviewer #1: Yes

Reviewer #2: Yes

Reviewer #3: No

6. Review Comments to the Author

Reviewer #1: The author has provided answers to concerns raised: sample size as well as the study location have been satisfactory stated.

Reviewer #2: I have accepted the response from the Authors and which to recommend the acceptance of the manuscript for publication.

Reviewer #3: After careful evaluation, I find that this manuscript currently falls short of the standards required for publication. The study presents an interesting concept, but there are significant concerns regarding clarity, methodology, and analysis which need to be addressed.

Abstract:

The results and conclusion sections of the abstract are rather confusing. It would be beneficial to restructure these sections for better clarity and coherence, ensuring that key findings and implications are presented in a straightforward manner.

Introduction:

The language used in the introduction is problematic, with notable issues of redundancy and unclear phrasing. A thorough revision to improve the language and eliminate repetitive statements is necessary to enhance the quality and readability of this section.

Methods:

The study exhibits a considerable loss to follow-up (33%, including lost data), which could introduce significant bias into the conclusions. It's crucial to address this issue and discuss its potential impact on the study's validity.

The decision to separate analyses by sex to reduce gender bias is noted. However, it seems that incorporating these variables into regression models could further minimize bias. Could you please explain the rationale behind not using such methods in your analysis?

Results:

The tables would benefit from including the number of patients involved in each specific analysis. This addition would greatly assist in understanding the scope and applicability of the findings.

Determining and specifying a cut-off value for each measurement under study would significantly enhance the interpretative value of the results.

Consideration should be given to including more variables in your models, such as sex, height, weight, BMI, age, etc., to provide a more comprehensive analysis.

7. PLOS authors have the option to publish the peer review history of their article (what does this mean?). If published, this will include your full peer review and any attached files.

Reviewer #1: **Yes: **Dr Ulu O Ulu

Reviewer #2: No

Reviewer #3: No

---

## [Author Response · Author response to Decision Letter 1]

18 Dec 2023

Response Letter

Manuscript ID number: PONE-D-23-24968

Title: Predictive value of anthropometric measurements in survival and free walking ability of geriatric hip fractures after surgery

Dear Editors and Reviewers,

Thanks very much for taking the time to review this manuscript. We appreciate all your generous comments and suggestions! Changes to our manuscript have been highlighted by using colored text in the revised manuscript. Please find our response below and my revisions in the re-submitted files.

Responses to editors

Thank you for your comments and tips concerning our manuscript. We would respond to reviewers and submit the revised manuscripts according to your prompts. Thank you. 

Responses to Reviewer 1

The author has provided answers to concerns raised: sample size as well as the study location have been satisfactory stated.

Response:

Thank you for your kind comments. I'm glad to hear that the concerns raised regarding the sample size and study location have been satisfactorily addressed.

Responses to Reviewer 2

The author has provided answers to concerns raised: sample size as well as the study location have been satisfactory stated.

Response:

Thank you for recommending the acceptance of the manuscript for publication. We are grateful for your support.

Responses to Reviewer 3

Abstract:

The results and conclusion sections of the abstract are rather confusing. It would be beneficial to restructure these sections for better clarity and coherence, ensuring that key findings and implications are presented in a straightforward manner.

Response:

Thank you for your valuable advice. The abstract has been revised as follows:

[Background: We aimed to explore the predictive value of anthropometric measurements in survival and free walking ability of geriatric hip fractures after surgery.

Methods: Eight common anthropometric measurements, including arm circumference (AC), waist circumference (WC), thigh circumference (TC), calf circumference (CC), biceps skinfold (BS), triceps skinfold (TS), suprailiac skinfold (SIS), and subscapular skinfold (SSS), were included to identify their predictive value in survival and free walking ability of geriatric hip fractures. The results of anthropometric measurements were compared between patients with different outcomes. Cox and logistics models were established to further identify the predictive value of anthropometric measurements.

Results: Comparison among groups indicated that individuals with different outcomes may have significantly different anthropometric measurements. In the Cox analyses based on all individuals, all models proved that the patients with higher AC, as well as CC and BS, may have a lower risk of 1-year mortality. Similarly, in the logistics analysis, AC, CC, and BS were proven to have strong predictive ability for 6-month and 1-year mortality in females and overall individuals. However, the predictive value of the eight common anthropometric measurements in free walking ability is not significant. 

Conclusion: AC, CC, and BS may have strong predictive ability for 6-month and 1-year mortality in all individuals and females.]

Introduction:

The language used in the introduction is problematic, with notable issues of redundancy and unclear phrasing. A thorough revision to improve the language and eliminate repetitive statements is necessary to enhance the quality and readability of this section.

Response:

Thank you for your valuable advice. The introduction part has been revised thoroughly with the help of native English speakers to enhance the quality and readability.

This part has been revised as follows:

[Hip fractures, occurring in the proximal femur, are commonly considered one of the most dangerous fractures in elder adults [1]. These fractures are primarily caused by low-energy trauma, often resulting from falls, while the root causes are age-related factors such as impaired balance, muscle weakness, and decreased bone mineral density [2]. Geriatric hip fractures are associated with high morbidity and mortality rates [3], primarily due to pre-existing comorbidities, surgical complications, and postoperative immobility [4]. Common complications include surgical site infections,

urinary tract infections, venous thromboembolism, pressure ulcers, pneumonia, and delirium. 

Nutrition status is a major risk factor for the outcomes of hip fractures and significantly impacts the rehabilitation of geriatric hip fractures [5]. Adequate nutrition is essential for bone regeneration, tissue repair, immune function, and wound healing, all of which may reduce the incidence of complications and improve the prognosis of geriatric hip fractures [6]. However, malnutrition as a common concern in older adults may contribute to poor healing, increased complications, prolonged recovery, low survival, and high disability following a hip fracture [7]. 

Assessing and addressing the nutritional needs in older adults with hip fractures is vital to improving their outcomes [5]. Anthropometric measurements, such as arm circumference, waist circumference, and others, can reflect an individual's nutritional status [8]. Poor anthropometric measurements are often associated with malnutrition, which can negatively impact the healing process and functional recovery in geriatric hip fracture patients. These anthropometric measurements provide valuable information about body composition and can assist healthcare professionals in assessing the nutritional needs of older adults with hip fractures. In this study, eight common anthropometric measurements, including arm circumference (AC), waist circumference (WC), thigh circumference (TC), calf circumference (CC), biceps skinfold (BS), triceps skinfold (TS), suprailiac skinfold (SIS), and subscapular skinfold (SSS), were included to identify their predictive value in survival and free walking ability of geriatric hip fractures.]

Methods:

The study exhibits a considerable loss to follow-up (33%, including lost data), which could introduce significant bias into the conclusions. It's crucial to address this issue and discuss its potential impact on the study's validity.

Response:

Thank you for your valuable advice. We noted that the high rate of loss to follow-up may cause bias in our study. However, we have tried our best to control the sample size in this study. Assuming that loss to follow-up is a random event, if the sample size is sufficient, the patients who are not lost to follow-up can adequately represent all patients. We believe that the bias introduced by loss to follow-up should also be controllable. Similarly, previous studies have also reported similar rates of loss to follow-up (PMID: 36510169, 36127624). More discussion about this issue has been added to the limitation part in our revised manuscript. 

The following part has been added to our revised manuscript:

[Firstly, as an observational study, this study had data loss and loss to follow-up, which may introduce bias into our findings. However, we have tried our best to control the sample size in this study. Assuming that loss to follow-up is a random event, if the sample size is sufficient, the patients who are not lost to follow-up can adequately represent all patients. We believe that the bias introduced by loss to follow-up should also be controllable.]

The decision to separate analyses by sex to reduce gender bias is noted. However, it seems that incorporating these variables into regression models could further minimize bias. Could you please explain the rationale behind not using such methods in your analysis?

Response:

Thank you for your valuable comments. We noticed that we forgot to state the specific variables for each model in the manuscript. In fact, in the Cox and logistic regression models, we have included the variables that showed significant differences in the univariate analysis. The included variables are age, sex (in models for all individuals only), BMI, and fracture type (femur neck and intertrochanteric). This variable selection approach is also commonly used in most literature, as it allows for the inclusion of relevant variables while ensuring model stability. 

We decided to conduct a gender subgroup analysis because we realized that differences in body measurements between different genders could lead to bias. Discrepancies in cutoff points, averages, and other measurements could potentially impact the outcomes. As a result, we chose to conduct a gender subgroup analysis to better understand how these differences may influence the results. 

More description of the variables in multivariable models has been added to our revised manuscript.

The following part has been added to our revised manuscript:

[Age, sex (in models for all individuals only), BMI, and fracture type were included in the Cox models 1 and 2 as co-variables.

Age, sex (in models for all individuals only), BMI, and fracture type were included in the multivariable logistics models 1 and 2 as co-variables.]

Results:

The tables would benefit from including the number of patients involved in each specific analysis. This addition would greatly assist in understanding the scope and applicability of the findings.

Response:

Thank you for your valuable advice. Added accordingly. Please see the tables in our revised manuscript.

Determining and specifying a cut-off value for each measurement under study would significantly enhance the interpretative value of the results.

Response:

Thank you for your valuable advice. We added an independent part for ROC curves to our study, and the detailed results of ROC analysis, including AUROC, 95%CI, potential cutoff values, and p values, have been added to our revised manuscript (Table 4 and supplementary table 11). Related descriptions have also been added to our revised manuscript.

The following parts have been added to our revised manuscript:

[Similar to Cox models, the logistics modes 1 and 2 were also established by adjusting for all types or single type of anthropometric measurements, respectively. ROC curves were established to evaluate the predictive values and to determine the potential cut-off values for each measurement. To avoid the bias caused by sex, the analyses were also performed in males and females separately.]

[Predictive values of anthropometric measurements

ROC curves were established to determine the predictive values and the potential cutoff values of anthropometric measurements for mortality and free walking ability. The multivariable analysis indicated the predictive ability of AC, CC, and BS, so we summarized their detailed results of ROC curves in Table 4. The detailed results of ROC analyses for all anthropometric measurements, including AUC, 95%CI, potential cutoff values, and p values, were summarized in Supplementary table 11. Similarly, the AC, CC, and BS showed significant predictive values for 1-year mortality in all individuals and females, while no significant results in males. For 1-year free walking ability, AC was proven to have significant predictive ability in all individuals and females, and CC was only proven in all individuals, while no significant results of BS were proven. The potential cutoff values of each anthropometric measurement have been summarized in Table 4 and supplementary table 11.

Table 4: Results of ROC analysis for AC, CC, and BS. 

All individuals

 1-year mortality 1-year free walking ability

 AUC(95% CI) Cutoff values p AUC(95% CI) Cutoff values p

AC 0.595 (0.529, 0.661) 21.88 0.002 0.568 (0.513, 0.623) 28.155 0.008

CC 0.597 (0.528, 0.667) 29.61 0.001 0.571 (0.514, 0.627) 29.625 0.007

BS 0.591 (0.526, 0.656) 13.005 0.003 0.523 (0.465, 0.581) 13.035 0.791

Males

 1-year mortality 1-year free walking ability

 AUC(95% CI) Cutoff values p AUC(95% CI) Cutoff values p

AC 0.556 (0.426, 0.687) 23.25 0.156 0.556 (0.451, 0.661) 26.295 0.87

CC 0.581 (0.459, 0.702) 29.59 0.074 0.664 (0.564, 0.763) 29.65 0.001

BS 0.600 (0.488, 0.712) 13.01 0.964 0.539 (0.440, 0.639) 13.165 0.786

Females

 1-year mortality 1-year free walking ability

 AUC(95% CI) Cutoff values p AUC(95% CI) Cutoff values p

AC 0.617 (0.542, 0.691) 28.245 0.002 0.575 (0.510, 0.640) 28.31 0.015

CC 0.608 (0.523, 0.694) 29.395 0.004 0.524 (0.455, 0.594) 23.19 0.76

BS 0.589 (0.508, 0.670) 12.98 0.014 0.517 (0.445, 0.589) 12.98 0.686

Note: AC: arm circumference, CC: calf circumference, BS: biceps skinfold.]

[Supplementary table 11: Detailed information of ROC curves; Supplementary table 12: Raw data of our study.]

Consideration should be given to including more variables in your models, such as sex, height, weight, BMI, age, etc., to provide a more comprehensive analysis.

Response:

Thank you for your valuable advice. We noticed that we forgot to state the specific variables for each model in the manuscript. In fact, in the Cox and logistic regression models, we have included the variables that showed significant differences in the univariate analysis. The included variables are age, sex (in models for all individuals only), BMI, and fracture type (femur neck and intertrochanteric). This variable selection approach is also commonly used in most literature, as it allows for the inclusion of relevant variables while ensuring model stability. More description of the variables in multivariable models has been added to our revised manuscript.

The following part has been added to our revised manuscript:

[Age, sex (in models for all individuals only), BMI, and fracture type were included in the Cox models 1 and 2 as co-variables.

Age, sex (in models for all individuals only), BMI, and fracture type were included in the multivariable logistics models 1 and 2 as co-variables.]

---

## [Decision Letter · Decision Letter 2]

26 Feb 2024

PONE-D-23-24968R2Predictive value of anthropometric measurements in survival and free walking ability of geriatric hip fractures after surgery.PLOS ONE

Dear Dr. Chu,

Thank you for submitting your manuscript to PLOS ONE. After careful consideration, we feel that it has merit but does not fully meet PLOS ONE’s publication criteria as it currently stands. Therefore, we invite you to submit a revised version of the manuscript that addresses the points raised during the review process.

We look forward to receiving your revised manuscript.

Kind regards,

Amir Human Hoveidaei

Academic Editor

PLOS ONE

Additional Editor Comments (if provided):

Dear authors

Many thanks for your submission

Based on the reviewer’s comments

I recommend major revision

Best,

AHH

Reviewers' comments:

Reviewer's Responses to Questions

**Comments to the Author**

1. If the authors have adequately addressed your comments raised in a previous round of review and you feel that this manuscript is now acceptable for publication, you may indicate that here to bypass the “Comments to the Author” section, enter your conflict of interest statement in the “Confidential to Editor” section, and submit your "Accept" recommendation.

Reviewer #4: All comments have been addressed

2. Is the manuscript technically sound, and do the data support the conclusions?

Reviewer #4: Yes

3. Has the statistical analysis been performed appropriately and rigorously? 

Reviewer #4: No

4. Have the authors made all data underlying the findings in their manuscript fully available?

Reviewer #4: Yes

5. Is the manuscript presented in an intelligible fashion and written in standard English?

Reviewer #4: Yes

6. Review Comments to the Author

Reviewer #4: I had the opportunity to review this interesting manuscript on the accuracy of common anthropometric measures to predict the outcome of hip fracture surgery in elderly population. The manuscript is well-written overall and employs a relatively sound methodology. I have some comments to enhance the methodology and quality of the presentation, which you can find below.

a. Line 93; Please further elaborate on how CCI is calculated and interpreted.

b. Please describe a summary of participants’ characteristics (at least their age and gender) in the first paragraph of the results section.

c. Table 1 may be somewhat misleading, and I recommend omitting it. Since males and females possess distinct anthropometric characteristics, aggregating them to assess the association between anthropometric measures and outcomes is not advisable. Tables 2 and 3 effectively convey the relationship between these measures and the outcome. I recommend a similar approach and omitting the aggregated section of Table 4, Figure 2, and Figure 3.

d. Please add a conclusion section at the end of the discussion section.

7. PLOS authors have the option to publish the peer review history of their article (what does this mean?). If published, this will include your full peer review and any attached files.

Reviewer #4: No

---

## [Author Response · Author response to Decision Letter 2]

28 Feb 2024

Response Letter

Manuscript ID number: PONE-D-23-24968

Title: Predictive value of anthropometric measurements in survival and free walking ability of geriatric hip fractures after surgery

Dear Editors and Reviewers,

Thanks very much for taking the time to review this manuscript. We appreciate all your generous comments and suggestions! Changes to our manuscript have been highlighted by using colored text in the revised manuscript. Please find our response below and my revisions in the re-submitted files.

Responses to editors

Thank you for your comments and tips concerning our manuscript. We would respond to reviewers and submit the revised manuscripts according to your prompts. Thank you. 

Responses to Reviewer 4

a. Line 93; Please further elaborate on how CCI is calculated and interpreted.

Response:

Thank you for your valuable advice. More description on CCI has been added to our revised manuscript.

The following part has been added to our revised manuscript:

[CCI is a widely used tool in medical research and clinical practice to quantify the burden of comorbidities in patients. The index takes into account 19 different conditions, such as heart disease, diabetes, and cancer, and assigns a score ranging from 1 to 6 to each condition based on its severity and impact on mortality [9]. The items and their weights were summarized in Supplementary table 1.

9. Charlson ME, Carrozzino D, Guidi J, Patierno C. Charlson Comorbidity Index: A Critical Review of Clinimetric Properties. Psychother Psychosom. 2022;91(1):8-35. Epub 20220106. doi: 10.1159/000521288.]

b. Please describe a summary of participants’ characteristics (at least their age and gender) in the first paragraph of the results section.

Response:

Thank you for your valuable advice. Added accordingly.

The following part has been added to our revised manuscript:

[The mean age of the study population was 72.38 ± 10.47 years, and the mean BMI was 21.77 ± 4.27 kg/m2. More female participants (374, 68.75%) were included in this study than males (170, 31.25%).]

c. Table 1 may be somewhat misleading, and I recommend omitting it. Since males and females possess distinct anthropometric characteristics, aggregating them to assess the association between anthropometric measures and outcomes is not advisable. Tables 2 and 3 effectively convey the relationship between these measures and the outcome. I recommend a similar approach and omitting the aggregated section of Table 4, Figure 2, and Figure 3.

Response:

Thank you for your valuable advice. These tables and figures have been revised accordingly. The updated Tables 1, 2, and 3 were the previous Tables 2, 3, and 4. The previous table 1 has been added to supplementary tables. 

d. Please add a conclusion section at the end of the discussion section.

Response:

Thank you for your valuable comments. Added accordingly.

The following part has been added to our revised manuscript:

[Conclusion

The predictive value of the eight common anthropometric measurements in elderly male patients after hip fracture surgery may not be significant in terms of mortality and rate of free walking. Additionally, their predictive role in free walking ability is also not significant in the both overall population and females. However, in all individuals and females, AC, CC, and BS have strong predictive ability for 6-month and 1-year mortality.]

---

## [Decision Letter · Decision Letter 3]

19 Apr 2024

Predictive value of anthropometric measurements in survival and free walking ability of geriatric hip fractures after surgery.

PONE-D-23-24968R3

Dear Dr. Chu,

We’re pleased to inform you that your manuscript has been judged scientifically suitable for publication and will be formally accepted for publication once it meets all outstanding technical requirements.

Kind regards,

Amir Human Hoveidaei, MD, MSc

Academic Editor

PLOS ONE

Additional Editor Comments (optional):

Dear authors,

Many thanks for your revisions.

I reviewed the paper as well as one of our peer-reviewers. All the comments are addressed. I recommend acceptance.

Best,

Amir H Hoveidaei, MD, MSc

Reviewers' comments:

Reviewer's Responses to Questions

**Comments to the Author**

1. If the authors have adequately addressed your comments raised in a previous round of review and you feel that this manuscript is now acceptable for publication, you may indicate that here to bypass the “Comments to the Author” section, enter your conflict of interest statement in the “Confidential to Editor” section, and submit your "Accept" recommendation.

Reviewer #4: All comments have been addressed

Reviewer #5: All comments have been addressed

2. Is the manuscript technically sound, and do the data support the conclusions?

Reviewer #4: Yes

Reviewer #5: Yes

3. Has the statistical analysis been performed appropriately and rigorously? 

Reviewer #4: Yes

Reviewer #5: Yes

4. Have the authors made all data underlying the findings in their manuscript fully available?

Reviewer #4: Yes

Reviewer #5: Yes

5. Is the manuscript presented in an intelligible fashion and written in standard English?

Reviewer #4: Yes

Reviewer #5: Yes

6. Review Comments to the Author

Reviewer #4: Thank you very much for revising the manuscript according to the comments. The manuscript's quality has improved after the revisions and I have no further comments.

Reviewer #5: Dear authors,

Many thanks for your submission.

All the comments are addressed and paper reads well now.

I recommend acceptance.

7. PLOS authors have the option to publish the peer review history of their article (what does this mean?). If published, this will include your full peer review and any attached files.

Reviewer #4: No

Reviewer #5: No

---

## [Editor Report · Acceptance letter]

30 Apr 2024

PONE-D-23-24968R3 

PLOS ONE

Dear Dr. Chu, 

I'm pleased to inform you that your manuscript has been deemed suitable for publication in PLOS ONE. Congratulations! Your manuscript is now being handed over to our production team.

Kind regards, 

on behalf of

Dr. Amir Human Hoveidaei 

Academic Editor

PLOS ONE